# NaYF$_4$: Yb,Er Upconversion Nanoparticles for Imaging: Effect on Red Blood Cells

Anna A. Doronkina [1], Vyacheslav I. Kochubey [1,2], Anastasia V. Maksutova [1], Alexander B. Pravdin [1], Artem M. Mylnikov [3], Nikita A. Navolokin [3,4] and Irina Yu. Yanina [1,2,*]

[1] Institution of Physics, Saratov State University, 410012 Saratov, Russia; annador95@bk.ru (A.A.D.); saratov_gu@mail.ru (V.I.K.); av.maksutova@gmail.com (A.V.M.); pravdinab@mail.ru (A.B.P.)

[2] Laboratory of Laser Molecular Imaging and Machine Learning, National Research Tomsk State University, 634050 Tomsk, Russia

[3] Department of Pathological Anatomy, Saratov State Medical University, 410012 Saratov, Russia; artyom-mylnikov@mail.ru (A.M.M.); nik-navolokin@yandex.ru (N.A.N.)

[4] Pathological Department, State Healthcare Institution "Saratov City Clinical Hospital No. 1 Named after Yu.Ya. Gordeev", 410017 Saratov, Russia

* Correspondence: irina-yanina@list.ru; Tel.: +7-8452-210716

**Abstract:** (1) Background: Upconversion nanoparticles (UCNPs) are a promising tool for biological tissue structure visualization and photodynamic therapy (PDT). The luminescence of such NPs is excited in the spectrum's near-infrared (NIR) region, while the NPs luminesce in the visible region. Conjugating such NPs with photodynamic dyes that absorb their luminescence makes it possible to increase the depth at which PDT is performed. (2) Methods: We conducted a comprehensive study on the possibility of using NaYF$_4$:Er:Yb UCNPs in vivo for imaging and PDT. The NPs were synthesized by a hydrothermal method. The synthesis of NPs with a size of 80 nm and hexagonal structure was demonstrated. (3) Results: The accumulation of NPs in organs after their intravenous injection into rats was studied. The effect of NPs on the shape, size, and degree of aggregation of red blood cells (RBC) was also investigated. (4) Conclusions: The possibility of luminescent visualization of NPs in histological sections and their subcutaneous distribution is demonstrated. All investigated particles showed moderate toxicity, causing mostly reversible changes.

**Keywords:** red blood cells; upconversionnanoparticles; luminescence; toxicity; laser; NIR





## 1. Introduction

In recent years, oncologic diseases have taken a leading position regarding human mortality. There are now many ways to treat these diseases. These include chemotherapy, surgery, radiotherapy, and photodynamic therapy (PDT). The latter uses visible or near-infrared (IR) radiation combined with a photosensitizer (PS) and oxygen [1]. There is also a method of plasmon resonance photothermotherapy (PPTT). This approach is based on the thermal destruction of biological tissues by laser heating of metal nanoparticles (NPs) placed inside the tissue, allowing operations to be performed at the microscale level [2].

PPTT efficiency largely depends on the means of control of the heating and denaturation process of biological tissues. There is a need for direct measurements of thermal fields inside a biological tissue in the immediate vicinity of plasmon resonance NPs. One possibility is the use of nanothermometers. These are NPs whose luminescence bands depend on the ambient temperature. Some promising nanothermometers are ZnCdS [3], CuInS$_2$ [4], and NaYF$_4$:Er:Yb [5–7].

Such NPs can also be used to visualize blood flow or areas of accumulation (tumors) simultaneously [8]. In this case, the luminescence of the NPs is usually recorded. One type of such a NP isupconversion NPs (UCNPs), e.g., NaYF$_4$:Er:Yb. They absorb excitation radiation in the infrared region of the spectrum and luminesce in the visible region [9]. The

excitation wavelength (975–980 nm) falls within the transparency window of biological tissues. This increases the penetration depth of the radiation that excites the luminescence. In addition, the conditions for PDT can be improved by creating complex NPs with target molecules and a photodynamic dye adsorbed to the surface [9]. The photodynamic dye absorbs the luminescence of the NPs. Thus, compared to the direct excitation of the dye in its absorption band, it is possible to act at a greater depth. One of the most promising materials for the development of PDT methods isfunctionalized UCNPs [10]. UCNPs can increase the therapy's efficacy and provide an additional photothermal effect while controlling the temperature of the treatment area [11–13]. For PDT, it is promising to prepare a UCNP (e.g., NaYF$_4$:Er,Yb) and then coat it with a PS-containing shell [14]. In this case, the efficiency of excitation of PS molecules is determined by the condition of the need to overlap the spectra of upconversion luminescence and the absorption of PS.

There are several widely used methods for the synthesis of UCNs [15], each of which has a number of unique features that determine its advantages and disadvantages.

The thermal decomposition method allows the synthesis of particles with a high degree of monodispersity with the possibility of particle size control in a relatively short reaction time [15–17]. This method's main disadvantage is maintaining a high reaction temperature (250–330 °C) in an oxygen-free environment. This places high demands on the equipment. Toxic wastes are also a major drawback [16]. In addition, the synthesized particles are usually stabilized by a surfactant. This poses difficulties in biological applications and requires surface modification [17].

The co-precipitation method is characterized by the absence of toxic by-products, by the synthesis temperatures that are not so high, and by its relative simplicity [15,16]. However, this method cannot obtain particles with a high degree of monodispersity. It should also be noted that the resulting particles contain a large amount of adsorbed water. This negatively affects their luminescence properties [15,16].

The sol–gel process is distinguished from other processes for synthesizing UCNs with higher yields [15,18]. The inability to control particle size and their significant aggregation are the main disadvantages of this method. Particle size is a critical parameter when using particles for biomedical applications. Particles and particle aggregates that are too large are difficult to remove from the body and can lead to capillary blockage. Particles that are too small have cellular permeability and, as a result, toxicity [19]. Therefore, to produce particles for biomedical applications, the sol–gel method is rarely used to synthesize UCNPs.

One of the most common synthesis methods is solvothermal synthesis [16]. The method is based on the high solubility of inorganic substances at elevated temperatures and pressures and the subsequent growth of crystals from the liquid phase. The necessary components of the system are a solvent, a mineralizer, and precursors for the grown crystals. The result of the use of organic solvents is the production of small, homogeneous particles. However, when the synthesis is done in organic solvents, the toxicity of the resulting substances increases the equipment requirements and the synthesis conditions.

The hydrothermal method, where water is the solvent, is a special case of solvothermal synthesis. This synthesis method is widely used due to its ease of implementation and low cost of starting materials. At the same time, carrying out the synthesis in an aqueous environment simplifies the process of surface modification for biomedical applications [15,20].

The hydrothermal method is promising for the production of ACC. It offers a wide range of possibilities for obtaining particles with specific parameters while controlling the influencing synthesis conditions. The scientific groups' data show that changing the hydrothermal synthesis conditions makes it possible to obtain particles of different morphologies, from rods to plates, and of different sizes, from 10 nm to 5 μm [21,22]. The hydrothermal method seems to be the most suitable because of the absence of toxic synthesis products and the possibility of preparing intensely luminescent particles of given sizes [23]. The issue of hydrothermal synthesis is discussed in more detail in the authors'paper [24].

The effectiveness of the use of NPs in oncological theranostics depends in part on the method of administration and delivery to the treated area. One of the injection methods is intra-arterial. It allows the delivery of agents in high concentrations to the mainstream. It usually takes less than a minute to pass through the entire circulatory system. The circulation's heterogeneous branching structure ensures good blood solution mixing within minutes, except for substances with a high first-pass metabolism. In this case, a larger amount of the drug will accumulate in the tumor and a smaller amount in the mononuclear phagocyte system. However, this route of injection is usually associated with a high risk. After intravenous injection, in vivo agents always interact with blood components, accumulating a large amount in the mononuclear phagocyte system (liver and spleen) [25].

Human erythrocytes are the most abundant blood cells. Due to their unique properties, such as biocompatibility, membrane flexibility, and high in vivo stability, represent a potential natural carrier system for various biologically active substances and contrast agents [26].

RBC-based drug delivery systems have evolved into diverse platforms for drugs and nanocarriers loaded into or onto the surface. This significantly alters the loaded drugs' pharmacokinetics, biodistribution, clearance, metabolism, activity, regulation, and pharmacodynamic properties, including their immunological and therapeutic properties. The use of RBCs as supercarriers opens up previously unrecognized opportunities for nanocarrier delivery into vascular cells, in some cases increasing the efficacy of nanocarrier targeting to the cells of interest [27,28].

In a review article, Mehrizi T.Z. [29] examined the data on the hemocompatibility of NPs with RBC from 2010 to 2020. The reviewed literature shows that negatively charged dendrimers, unsaturated/uncharged liposomes, pegylated NPs, and RBC can improve RBC quality and hemocompatibility. However, large cationic dendrimers, liposomes composed of saturated long acyl chain lipids, and cationic chitosan NPs are less compatible with RBC. In addition, these polymeric NPs allow for more surface modifications and manipulations, making them more hemocompatible. Among metal NPs, gold and iron NPs have reasonable compatibility with RBC. However, the smaller size, higher concentration, and longer exposure time of these NPs may induce hemolysis and morphological changes in RBC.

When NPs larger than 50 nm are used, membrane interaction is the main mechanism of interaction with RBC. This changes the RBC's shape and size and the aggregation degree. In numerous and varied experiments with blood samples, it has been shown that the degree and rate of erythrocyte aggregation depend not only on the concentration of the cells and the state of their surface but also on the physicochemical properties and concentration of blood plasma proteins [30,31].

However, one must ensure that the binding of NPs to the RBC membrane does not lead to disastrous morphological and physiological consequences. Determining the effect on blood cells is necessary before injecting the drug into the bloodstream.

The aim of this work is to create a NP preparation, which is a UCNP, on the surface of which address molecules and a photodynamic dye are added; to inject a suspension of NPs to experimental animals and observe the distribution in organs; to observe of changes of the properties of the erythrocyte membrane in vitro and in vivo under the action of these NPs with different surface conditions. That is extensive research on nanoparticle effects on red blood cells and histological analysis of organs and tissues.

## 2. Materials and Methods

### 2.1. Synthesis and Size Characterisation of UCNPs

For the synthesis of NPs, we used $Re_2O_3$ (Re—Y, Yb, Er)—Rare Metals Plant, Novosibirsk, Russia; sodium citrate 5,5-hydrate $Na_3C_6H_5O_7*5,5H_2O$—LenReactiv, Saint-Petersburg, Russia; ammonium fluoride NH4F—LenReactiv, Saint-Petersburg, Russia.

The UCNPs were synthesized by our variant of the hydrothermal method [24] according to the technology developed by us (Supplementary File S1). In this case, an increased

concentration of citric acid was used, but the amount of sodium in the citrate composition was stoichiometric. No other sources of sodium were used.

Newly synthesized NPs, including those annealed at a temperature of 500 °C, were used for the research. In addition, part of the NPs were coated with $SiO_2$ shells according to the standard procedure [32].

We also generated a rigid protein corona of albumin on the surface of the NPs (Supplementary File S1).

Coating the surface of NPs with a protein corona allows the attachment of dyes. We used human serum albumin (HSA) labeled with methylene blue dye [33], which covalently binds to the free amino groups of the protein.

A dye solution (0.1%, 100 μL) was added to the suspension of coated NPs. The mixture was kept under stirring on a PTR-25 mini-rotator for 3 h. The procedure of precipitation and washing of NPs was then repeated. The resulting NPs in the buffer were stored in a refrigerator at 4 °C.

The size and shape of the NPs were determined using a MIRA 2 LMU scanning electron microscope (TESCAN, Brno, Czech Republic). The structure of the crystal lattice was determined using an ARLX'TRA X-ray diffraction phase analyzer (Tharmo Fisher Scientific, Waltham, MA, USA).

### 2.2. Animal Experiments

The present study was conducted in adult white Wistar rats. Work with laboratory animals was performed in accordance with the research protocol, which does not conflict with the requirements of the "International Guiding Principles for Biomedical Research Involving Animals" (December 2012, Council for the International Organization of Medical Sciences andThe International Council for Laboratory Animal Science). Experimental work in vivo was performed at the Center for Collective Use (CCU) of the Research Institute of Fundamental and Clinical Uronephrology of Saratov State Medical University.

In this study, the development of alveolar liver cancer (cholangiocarcinoma, PC1) was modeled by injecting 0.5 mL of a 25% tumor suspension in Hanks'solution subcutaneously in the scapular region. All animals underwent the necessary quarantine in the vivarium, where they were housed in individual cages in an enclosed, heated room at a temperature of 20–25 °C in individual cages. The dietary regimen was standard, using compound rodent chow. The experiment was performed on day 28 after tumor implantation. The subject and the description of the experiments were approved by the Ethics Committee of the Saratov State Medical University (Protocol No. 13, 3 May 2011).

Rats were intravenously injected with a suspension of NaYF4 NPs with different surfaces (annealed, unannealed, with shells of $SiO_2$, albumin, or albumin with a dye). The drug concentration in physiological saline ranged from 0.5 to 2 mg/mL. The volume of drug injected into the rats was, on average, 0.45–0.6 mL. The drug dose for intravenous injection was 3 mg/kg. To analyze the photodynamic effect of NPs, animals with cancerous tumors were irradiated one day after NP administration. The animals were collected for histological examination and one day later for histological examination and blood sampling for visual assessment of RBC status.

As a control, samples were taken before the injection of NPs and also before irradiation.

For histological examination, tumor and internal organ specimens were fixed in 10% neutral buffered formalin, subjected to standard ethanol (isopropyl alcohol) processing, and embedded in paraffin blocks. To describe changes in the tumor and organs, 3–5 μm thick sections were stained with hematoxylin and eosin. Sections of 5 μm thickness from paraffin blocks of rat organs and tumors were used as study objects. The sections were cut on an Accu-Cut SRM microtome (Sakura, Japan). The sections were then spread in a water bath, mounted on glass slides, and deparaffinized with Bio-clear solution (Bio-Optica, Milan, Italy). Dewaxing was performed at 22 °C for 30 min.

Morphometric measurements and microphotography were performed on at least 10 fields of view of each micropreparation using the µVizo-103 Medical Transmitted Light Microvisor (LOMO, Sankt-Peterburg, Russia).

A video system based on a Dhyana 400DC video camera (Tuscen Photonics, Fuzhou, China) was developed to record luminescence images of sections. Interchangeable interference filters (Alluxa Inc., Santa Rosa, CA, USA) were used to select the luminescence wavelength selectively. The filters allow the selection of the ranges 521.5 ± 17 nm and 546.5 ± 5 nm. The sensitivity of the system allows luminescence images to be taken in a narrow range, providing data not only on the spatial distribution of NPs in an object but also on the spatial distribution of temperature. The luminescence was excited usinga 980 nm laser diode STLE-M-980-W010 (Thorlabs, Newton, NJ, USA) with fiber output. The radiation was collimated to a spot with a diameter of 1 cm. The radiant power could be varied from 0.1 to 1 W. The same source was used to irradiate laboratory animals. The tumor site was irradiated for 30 min at a dose of 1 mW/cm$^2$.

### 2.3. RBC Studies

Microscopic observations of the dynamics of shape and size of laboratory rat erythrocytes (and their aggregates) in contact with a suspension of NPs were performed using a LOMO Mikmed-2 microscope (LOMO, Sankt-Peterburg, Russia). The available Videoscan-415/P camera (Videoscan, Moscow, Russia) was used for image acquisition.

To detect and evaluate the size of RBCs and their aggregates in microphotographs, we used a freely available image processing package—ImageJ2 distribution kit with "batteries included", which combines many plug-ins to facilitate scientific image analysis.

To obtain an in vitro mixed suspension with a given concentration of cells and NPs, we used a suspension in a physiological solution of synthesized NaYF$_4$:Yb,Er NPs with an average size of 80 nm (Figure 1). Four types of NPs were used:

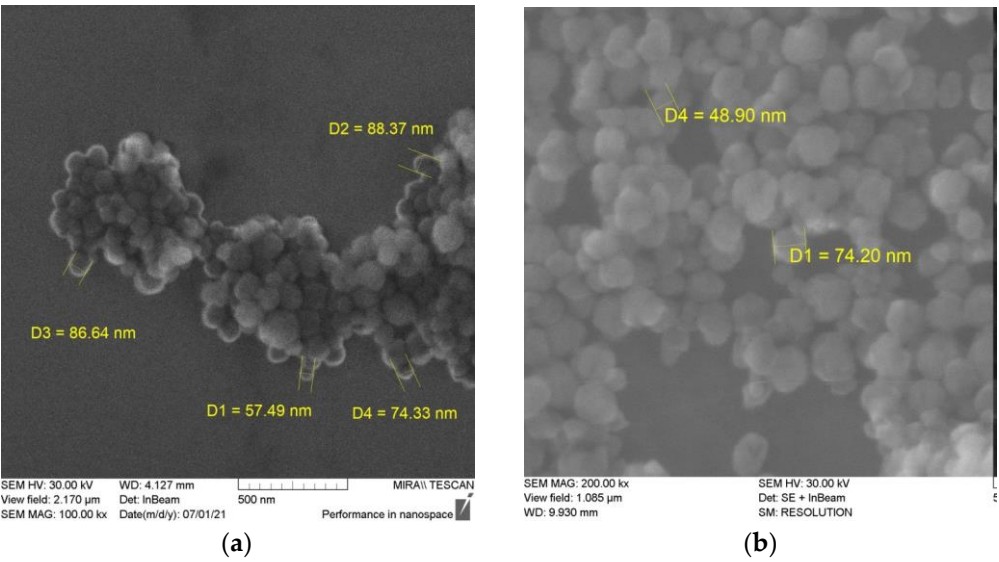

(a)                    (b)

**Figure 1.** SEM images of NaYF$_4$: Yb$^{3+}$, Er$^{3+}$ particles coated with SiO$_2$ shell. Magnification 30× (**a**) and 200× (**b**).

Type 1.—Synthesized NaYF$_4$ particles "as is", coated with citrate anions as the synthesis was performed out using sodium citrate;

Type 2.—NaYF$_4$ particles annealed at a temperature of 500 °C to remove citrate anions from the surface [24];

Type 3.—NaYF$_4$ particles coated with SiO$_2$ [34];

Type 4.—NaYF$_4$ particles coated with HSA molecules and Methylene Blue.

The study of the interaction of NPs with the red blood cell membrane was performed on laboratory rat red blood cells washed from plasma. Heparin-stabilized blood collected

from a laboratory animal was diluted with twice the volume of saline and centrifuged in an OP-8 laboratory centrifuge at 2000 rpm for 5 min. The supernatant was discarded, and the red cells were washed twice with saline and centrifuged. After the last centrifugation, 20 µL of the red cell mass was carefully removed from the bottom of the tube with a pipette without stirring and added to the tube containing 3.98 mL of saline.

Four types of samples were prepared from the resulting 0.5% RBC suspension for in vitro interaction studies:

(a)    0.25% RBC suspension in saline;
(b)    0.25% RBC suspension with NPs $NaYF_4$ (0.1 mg/mL);
(c)    0.25% RBC suspension with NPs $NaYF_4$ (0.2 mg/mL);
(d)    0.25% RBC suspension with NPs $NaYF_4$ (0.4 mg/mL).

The resulting suspensions were incubated in two experiments in a TZh-TC-01 liquid thermostatic bath at 39 °C and 25 °C.

Microscopic control samples were taken every 40 min for 2 h. In addition, for room temperature experiments, measurements were taken after one day of incubation.

For measurements, a drop of the stirred suspension was placed in a 0.17 mm thick, 5 mm wide, two-sided, removable glass cuvette. For the in vivo experiment, heparin-stabilized blood collected from a laboratory animal was diluted with twice the volume of saline and washed in a centrifuge for 5 min at 2000 rpm. After centrifugation, 0.02 mL of the erythrocyte mass was removed from the bottom of the tube and added to a tube containing 1.8 mL of saline. In addition, 0.5 mL of physiological solution was added to 0.5 mL of the previously obtained erythrocyte suspension to obtain a working concentration of erythrocytes.

The resulting images were processed using ImageJ.

The following indicators were used to characterize the size and shape of the blood cells:

(1)    Area is the projection area;
(2)    Feret is the maximum diameter of Feret. It corresponds to the smallest diameter of the circle circumscribed around the object, i.e., the greatest length of the object. Instrumentally, this length is obtained by measuring the length with a caliper, and in pomology, it is called the largest linear diameter, longitudinal diameter, i.e., the diameter of a circle described around cells of any shape.

## 3. Results and Discussion

### 3.1. NPCharacterization

According to our electron microscopy data, by varying the citrate concentration within small limits, it is possible to vary the size of the NPs from 26 to 200 nm. Particles larger than 100 nm are well facetted, while smaller particles are rounded. We chose a synthesis for the experiment that yielded NPs with an average size of 80 nm (Figure 1, Supplementary File S1).

The NPs had a hexagonal structure (Supplementary File S1).

Figure 2 shows the luminescence spectrum of NPs without a shell and with a shell of albumin with a dye under laser excitation (980 nm). The figure shows the luminescence peak of the dye in the region of 670 nm. This confirms both the presence of the dye in the shell and the possibility of its excitation by infrared radiation.

The erbium radiative transitions that form the luminescence spectrum are shown in Table 1. The lines show a fine structure due to the interaction with the hexagonal lattice of the NPs. The relative decrease in intensity of the red radiation is explained by its greater absorption by Methylene Blue, which is located in the shell of the nanoparticles.

**Table 1.** Erbium radiative transitions.

| 522 nm | 541 nm | 658 nm |
|---|---|---|
| 2H11/2 (Er) $\rightarrow$ 4I15/2 (Er) | 4S3/2 (Er) $\rightarrow$ 4I15/2 (Er) | 4F9/2 (Er) $\rightarrow$ 4I15/2 (Er) |

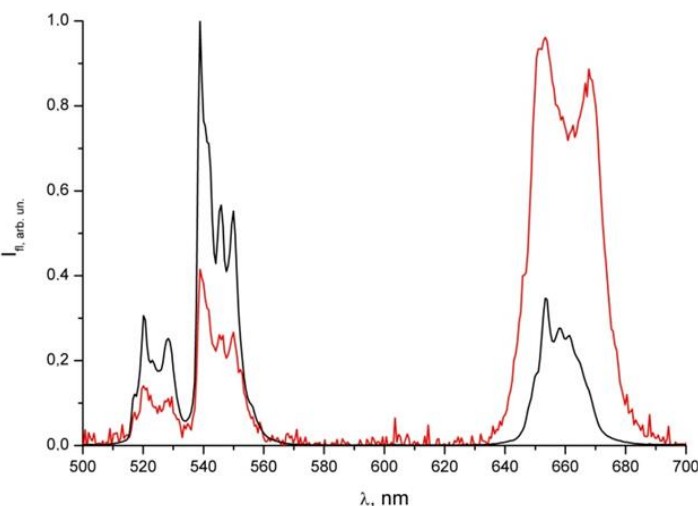

**Figure 2.** Maximum-normalized luminescence spectra of UCNPs $NaYF_4$: $Yb^{3+}$, $Er^{3+}$ (black line), and UCNPs coated with Methylene Blue (red line).

### 3.2. Imaging

When NPs are injected into the tail vein of a rat, luminescence is observed at the injection site (Figure 3). In this case, the transverse size of the excitation radiation is 0.5 cm, and the radiation power is 200 mW. However, despite the accumulation of NPs in the tumor, luminescence is observed when the excitation radiation is focused into a beam with a diameter of 2 mm, and the power is increased to 1 W. This is due to the deep location of the tumor. With the described excitation mode, registration is possible with weak scattered light, but a halo is recorded around the irradiation site due to light scattering in the tumor and skin over the tumor (Figure 4a). When NPs are injected directly into the tumor, a diffuse image is observed due to both the distribution of NPs over the tumor and scattering within the tumor (Figure 4b).

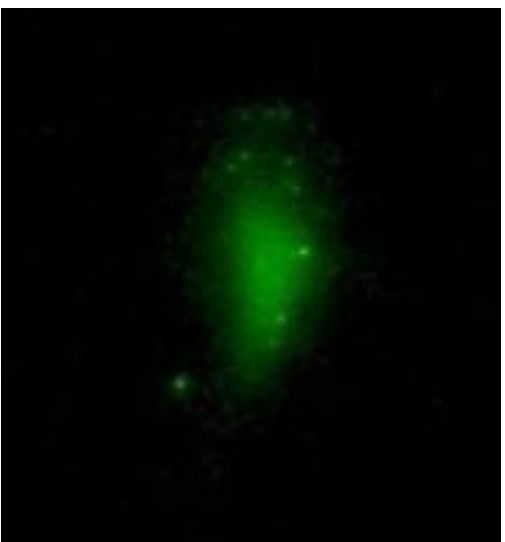

**Figure 3.** Luminescent image of the site of injection of NPs into the tail vein. NPs are displayed as single points of light or as a cluster of points, the size of which is determined by the size of the diffraction spot formed by the optical system.

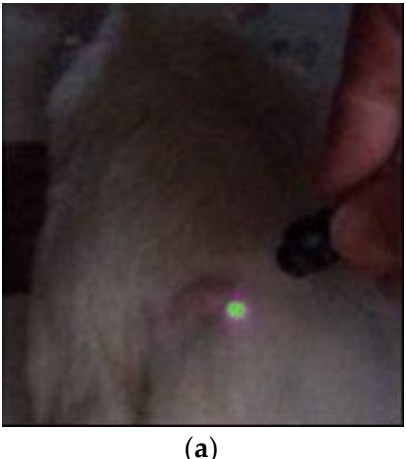 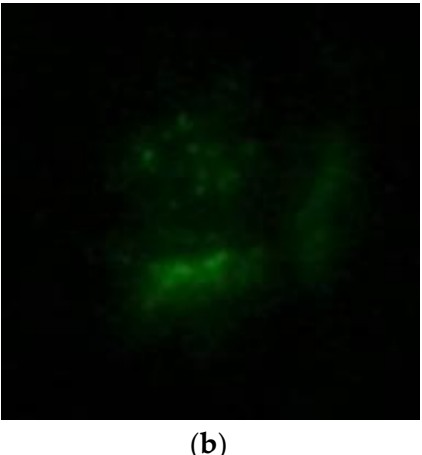

(**a**) (**b**)

**Figure 4.** Luminescence of NPs: (**a**)—in a tumor with intense excitation. Stimulating luminescence of NPs in a tumor by IR laser irradiation; (**b**)—injected directly into the tumor. NPs are displayed as single points of light or as a cluster of points, the size of which is determined by the size of the diffraction spot formed by the optical system.

To localise the NPs precisely, it is necessary to image where the NPs lie within the tumor. This allows the NPs to be irradiated directly in the tumor area without irradiating neighboring areas. We did not perform the visualization in a point-by-point mode but by photoregistration of the entire image within the irradiation area (1 cm).

The video system we developed also recorded luminescence images in histological sections (Figure 5).

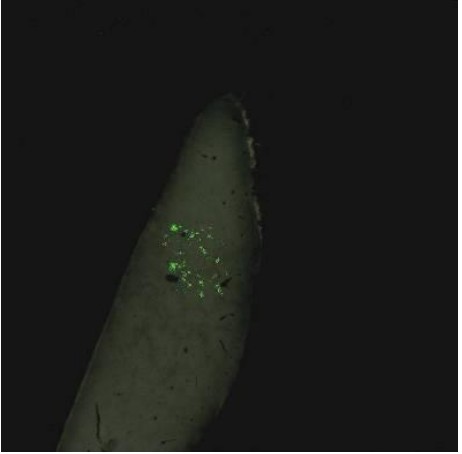

**Figure 5.** Spatial distribution of luminescent NPs in a histological section of the liver. NPs are displayed as single points of light or as a cluster of points, the size of which is determined by the size of the diffraction spot formed by the optical system.

*3.3. Histology*

Histological studies have shown (Supplementary File S2) that all the particles studied have moderate toxic properties, causing mostly reversible changes, with the least changes occurring when type 3 ($NaYF_4/SiO_2$) NPs are administered to rats at a dose of 0.5 mg/kg, and there are no changes in the myocardium, lungs, kidneys, and liver, organs corresponding to the norm. Changes in the spleen can be analyzed from the point of view of the immune response to the introduction of nanoparticles. No significant changes were observed. Thus, according to the morphotoxicological study of the described particles, the $NaYF_4/SiO_2$ NPs are the safest and most promising due to the lowest toxicity. When

testing for antitumor activity, other particles have a greater anticancer effect;they can be considered antitumor agents because they are not hypertoxic.

Pathologic anatomy, by definition, distinguishes between reversible changes (impaired perfusion, edema, infiltration of blood cells) and irreversible changes (necrosis, apoptosis) [35]. Therefore, when describing histologic changes, we conclude that the agents under study are moderately toxic if we observe signs of impaired blood supply, moderate cell damage, and the absence of necrosis of organ cells.

Necrosis in the tumor was observed in all groups, but the percentage of the necrotic area to the total area of the sections was different. For type 1 particles, it is 70%, type 2—60%, type 3—60%, and type 4—80%. In the control group, the changes in the organs were similar but less pronounced, which may be due to a slight necrosis of the tumor tissue (up to 10%).

The most pronounced changes in tumor tissue were observed after double injection (particle concentration 1 mg/mL, dose 10–13 mg/kg) and IR laser irradiation for 30 min (980 nm, power density was 500 mW/cm$^2$). At the triple dose (dose 15–20 mg/kg), the tumor is preserved, but the organs also have a structure close to the norm, possibly due to the accumulation of particles in the tumor tissue. However, irradiation did not cause tumor necrosis or intoxication; the particles remained in the tumor and did not damage organs, tissues, or erythrocytes.

Changes in the introduction of particles are reversible and not serious, but erythrocyte aggregation occurs, which correlates with changes in the blood filling of organs and the development of hemosiderosis. After two or three injections, a pigment appears in the epithelium of the kidneys, indicating possible excretion in the urine. Large amounts also accumulate in the white and red pulp of the spleen.

### 3.4. Studies on the State of the RBCs

In the course of the experimental work, the formation of aggregates by RBCs interacting with UCNPs was assumed. In order to evaluate the change in the shape of the RBC and the formation of aggregates, a visual analysis of the samples taken was performed using microphotographs taken with a light microscope (Figure 6).

Microphotographs of erythrocytes are also shown in the Supplementary File S3.

RBC aggregation was observed in all samples.

In in vitro studies, the size of aggregates (the degree of membrane modification by NPs) increases in the series of NP types 1–3. The smallest aggregates are formed when using NPs with a surface coated with HSA, while the concentration dependence of the size of the aggregates is practically not observed.

In vivostudies show somewhat different results.

In type 1 samples without irradiation, there is a high degree of aggregation but no change in shape, as shown by the calculation of the mean diameter of the RBC.

In type 2 samples without irradiation, echinocytes are formed, which form aggregates, and the mean diameter of the RBC is greater than in the control.

In type 3 samples without irradiation, aggregation is observed, but free red cells are present.

In type 1 samples, a change in shape and the formation of echinocytes are observed after irradiation, indicating a violation of the RBC cytoskeleton.

In samples with type 2 NPs after irradiation, the shape of the erythrocytes is close to that of the control group.

In type 3 samples, a change in the shape of the RBCs is also observed after irradiation, and rare echinocytes are found. A slight aggregation may be observed.

The results of the in vivo experiments are shown in Figure 7 and Supplementary File S3.

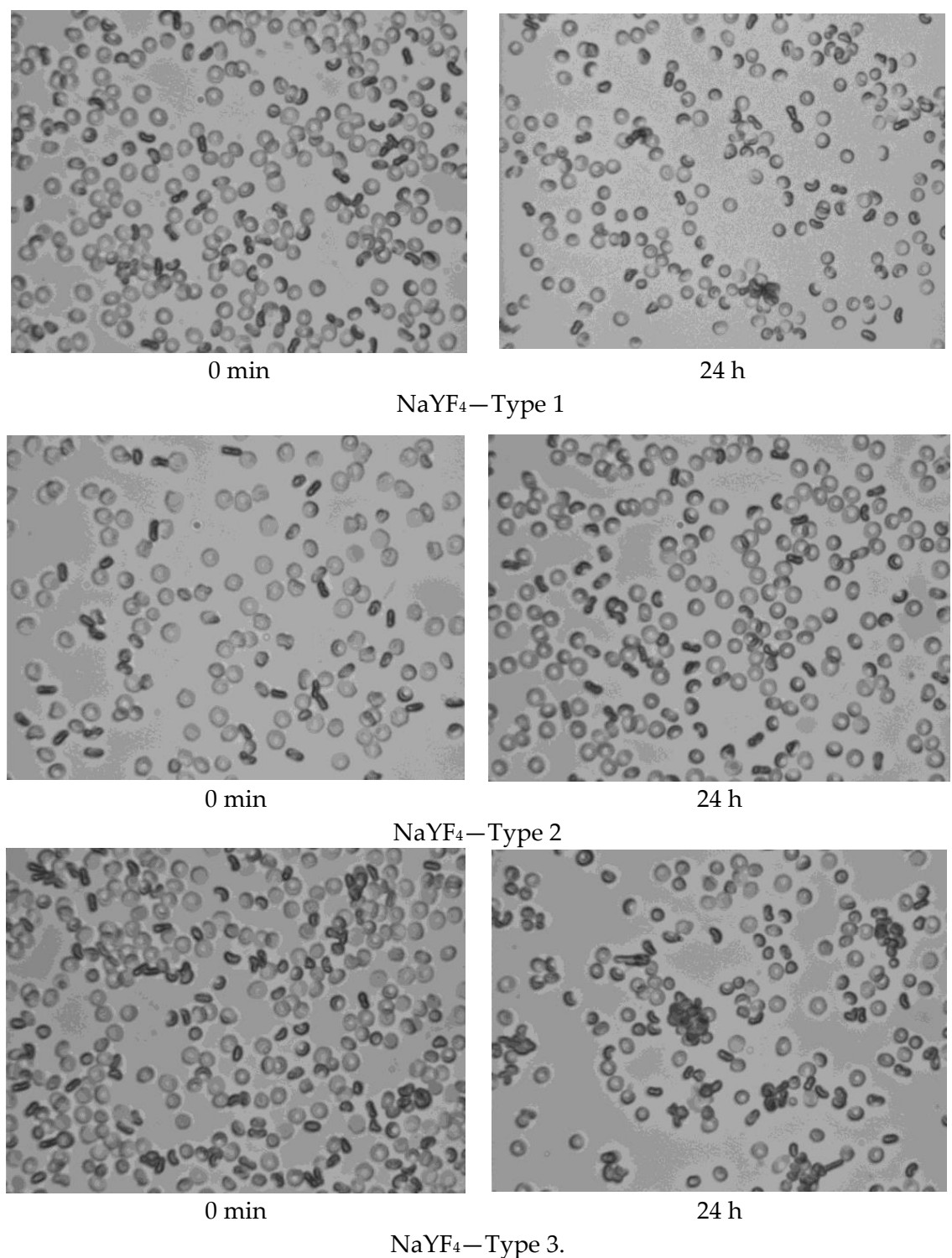

**Figure 6.** Microphotographs of a 0.25% suspension of RBC in NaYF$_4$ solution (0.1 mg/mL). RBC shape change and aggregate formation are observed.

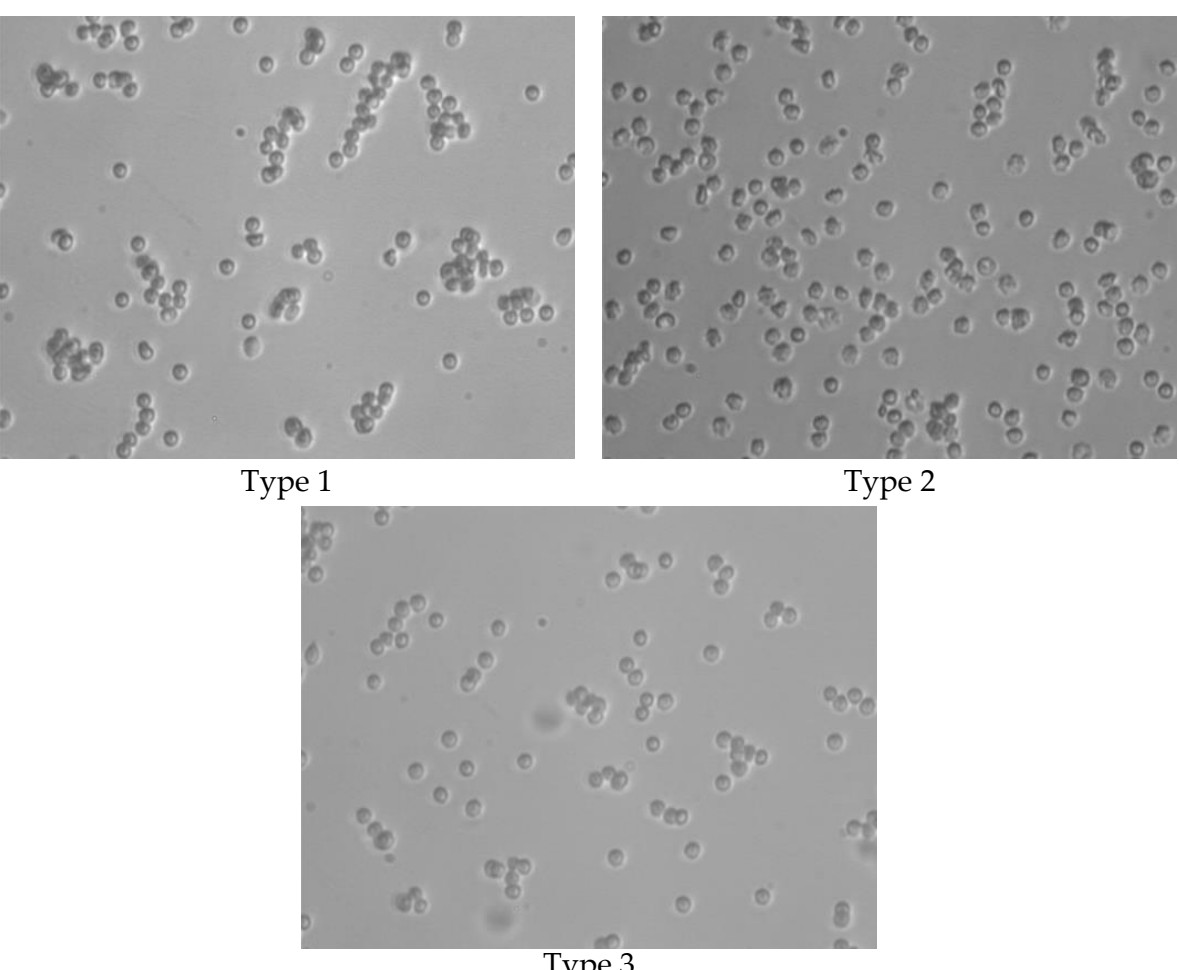

Type 1

Type 2

Type 3

**Figure 7.** Microphotographs of a 0.25% suspension of RBCwere obtained one day after the introduction of a suspension of NPs NaYF$_4$ (1 mg/mL). Type 1: high degree of aggregation but no change in shape; Type 2: echinocytes are formed, which form aggregates. The mean diameter of the RBCs is larger than in the control; in Type 3, aggregation is observed, but free red cells are present.

It can be noted that the aggregate area sizes are comparable in in vivo experiments and in samples incubated with an NP concentration of 0.1 mg/mL. The closeness of the values is due to the fact that when a suspension of NPs at a concentration of 1 mg/mL is introduced into the bloodstream of a rat, the concentration of NPs is distributed throughout the volume of the animal.

The interaction of NaYF$_4$ NPs with the RBC membrane is accompanied by an increased formation of aggregates, indicating a modification of the properties of the cell membrane surface. The size of the aggregates formed depends strongly on the type of surface of the NPs. It is most likely that the changes occur as a result of a phase transition of the membrane, which causes a change in its mechanical properties. Such changes lead to a change in the sphericity coefficient of RBCs [36].

The results of the calculation of the average area of the aggregates and the average size of the RBC are shown in Figures 8 and 9 and in Supplementary File S3.

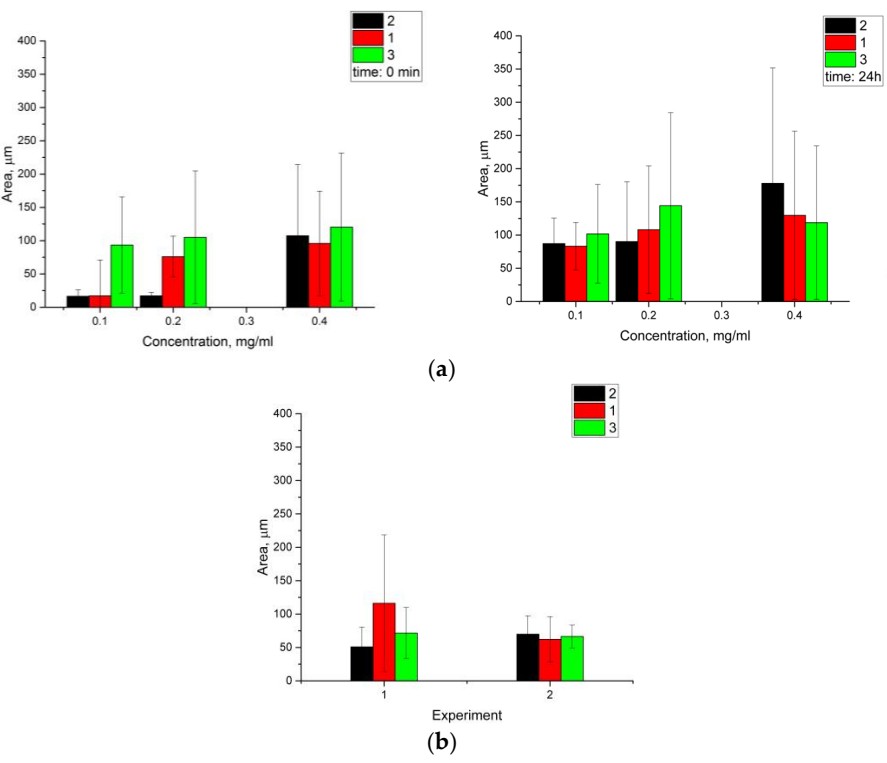

(**a**)

(**b**)

**Figure 8.** Average area of RBC for experiments (**a**)—in vitro (for different times), (**b**)—in vivo. Designations in Figures: 1—type 1, 2—type 2, 3—type 3. (**b**): Experiment 1—rat without irradiation, experiment 2—rat after irradiation.

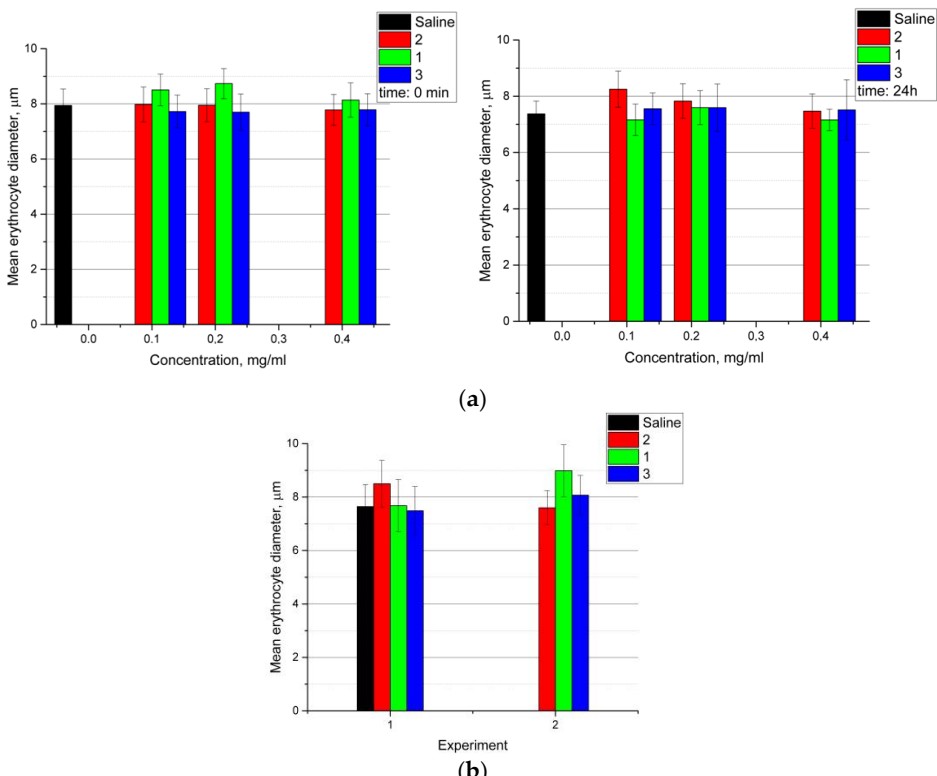

(**a**)

(**b**)

**Figure 9.** Mean diameter of RBC for experiments (**a**)—in vitro (for different times), (**b**)—in vivo. Designations in the figures: saline—control, 1—type 1, 2—type 2, 3—type 3. (**b**): Experiment 1—rat without irradiation, experiment 2—rat after irradiation.

The dynamics of aggregate formation show a concentration dependence: at high concentrations of NPs;large aggregates are formed immediately; at low concentrations, the size of the aggregates increases with the incubation time. As the concentration of NPs increases, an increase in the formation of aggregates is observed, as well as an increase in the value of the average area of the aggregates.

We believe that the change in RBC size is due to the appearance of echinocytes. The change in the RBC size and the appearance of echinocytes during the interaction of RBCs with NPs is related to the area of penetration of the NPs into the membrane: into the outer or inner layer. In our case, it is most likely that the NPs are embedded in the outer layer of the membrane, which can lead to structural changes due to the redistribution of lipids in the membrane. Incorporating the NPs into the membrane, with its subsequent stretching, leads to the appearance of protrusions and the formation of echinocytes.

Due to the change in shape of RBCs to stellate, the size of the described circle increases. Accordingly, the average size of RBCs incubated with NPs is larger than that of RBCs in the control group. However, there is also a decrease in the size of the diameter, which may be due to the shape of the RBC: the larger and sharper the rays, the smaller the size of the described circle around the cells. The change in diameter is probably due to a violation of the cytoskeleton or to the introduction of NPs into the RBC membrane.

The effect of all types of NPs leads to a slight change in the diameter of the RBC, while a suspension of high-concentration NPs leads to a decrease in size immediately at the moment of introduction into the RBC sample, while at low particle concentrations, even a slight increase in size is observed at the beginning of the experiment.

The change in RBC size is rather weak depending on the type of NP surface, although this weak dependence is traceable for all concentrations of NPs in the sample. The largest decrease in RBC diameter occurs with type 1 NPs.

The observed changes in the shape of erythrocytes and the appearance of echinocytes contribute to the observed increase in aggregation. However, it was not possible in our study to assess the effect on erythrocyte functionality or lifespan, but there was no increase in erythrocyte destruction (hemolysis was not detected).

We will consider the results obtained when conducting research on the effect of photodynamic exposure on tumor growth and organ condition with repeated administration of NPs.

## 4. Conclusions

An analysis of the results obtained allows us to draw the following conclusions.

NPs with additional shells or functionalized by coating the surface with photoactive molecules are considered, allowing the creation of particles with multiple modalities. The phototoxicity of such particles is considered separately. When using NPs for therapy or diagnostics of the state of living objects, the issue of toxicity is relevant. The toxic effect of UCNPs on the body depends on their concentration at the time of administration as well as on the total number of NPs in relation to the body weight. From the concentration dependencies considered, based on the results of histological and biochemical studies, it has been shown that such particles usually do not show significant toxicity and that the maximum allowable concentration of particles can be considered to be 2 mg/mL. Based on the analysis of histological material, the following changes were observed in the organs of the animals. Changes in the liver and lungs were reversible. Myocardial edema developed in the heart.

The phototoxicity of these particles was also demonstrated in animals bearing a transplanted liver tumor. The most pronounced antitumor effect was achieved with a double injection (particle concentration 2 mg/mL, dose 11.5 mg/kg) and IR laser irradiation of the tumor for 30 min (980 nm, power density 1 mW/cm$^2$). The presumed route of excretion of the particles and the body is urine. The accumulation of particles occurs in the spleen, which agrees with other researchers' data.

We have shown that it is possible to obtain luminescent images of the localization site of the nanoparticles we use by excitation of the luminescent region with a broad beam and

photoregistration. The possibility of imaging depends, in particular, on the quality of the synthesized nanoparticles and, in particular, on the quantum yield of luminescence. In order to locate the nanoparticles precisely, it is necessary to image where the nanoparticles are located within the tumor. This allows the nanoparticles to be irradiated directly in the tumor area without irradiating neighboring areas. We did not perform the visualization in a point-by-point mode but by photoregistering the entire image within the irradiation area (1 cm).

Laser irradiation of tumors in PC-1 liver cancer animals resulted in liver detoxification. The main effect on the kidneys after particle administration was the collapse of the space between the vascular glomerulus and the Shumlyansky–Bowman capsule. In the spleen, there was an increase in pigment in the red pulp. With two and three injections, a pigment appeared in the epithelium of the kidneys, indicating its possible excretion with the urine. Large amounts of accumulation also occurred in the white and red pulps of the spleen.

The interaction of $NaYF_4$ NPs with the erythrocyte membrane is accompanied by an increased formation of aggregates, indicating a modification of the properties of the cell membrane surface. The size of the aggregates formed is strongly dependent on the type of surface of the NPs. The dynamics of aggregate formation show a concentration dependence: at high concentrations of NPs, large aggregates form immediately; at low concentrations, the size of the aggregates increases with incubation time.

Exposure of erythrocytes to all types of NPs results in a slight decrease in erythrocyte diameter, with the change in erythrocyte size being rather weakly dependent on the type of NP surface.

The results show that yttrium sodium fluoride NPs with a surface coated with human serum albumin have better biocompatibility with RBC. This type of nanoparticle may be more effective in imaging and PDT for superficial tumors with developed blood vessels. More particles will enter and accumulate. For example, hemangiomas, skin tumors, glioblastomas, meningiomas

**Supplementary Materials:** The following supporting information was downloaded from https://www.mdpi.com/article/10.3390/photonics10121386/s1, S1: Synthesis and characterization of nanoparticles; S2: Results of histological studies of organs; S3: Analysis of red blood cell size and shape.

**Author Contributions:** Conceptualization, V.I.K. and I.Y.Y.; investigation, A.A.D., V.I.K., A.V.M., A.B.P., A.M.M., N.A.N. and I.Y.Y.; methodology, V.I.K. and I.Y.Y.; supervision, V.I.K. and I.Y.Y.; visualization, V.I.K.; writing–original draft, V.I.K. and I.Y.Y.; writing–review andediting, V.I.K. and I.Y.Y. All authors have read and agreed to the published version of the manuscript.

**Funding:** The study was supported by a grant from the Russian Science Foundation No. 21-72-10057, https://rscf.ru/project/21-72-10057/ (accessed on 1 December 2023).

**Institutional Review Board Statement:** The animal study protocol was approved by the Ethics Committee of Saratov State Medical University (protocol code 01 and 5 September 2023).

**Informed Consent Statement:** Not applicable.

**Data Availability Statement:** The data presented in this study are available uponrequest from the corresponding author. The data are not publicly available due to privacy or ethical restrictions.

**Conflicts of Interest:** The authors declare no conflict of interest.

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
