# Peer review of "NaYF4: Yb,Er Upconversion Nanoparticles for Imaging: Effect on Red Blood Cells"

_photonics, doi:10.3390/photonics10121386_

Round 1

Reviewer 1 Report

Comments and Suggestions for Authors

1.        Could the authors provide more detail on the hydrothermal method used for the synthesis of the NPs? Specifically, how does this method compare to other synthesis methods in terms of particle size uniformity and surface properties?

2.        The manuscript mentions moderate toxicity of the NPs. Could the authors elaborate on the criteria used to determine this level of toxicity?

3.        Is there any long-term study planned or conducted to evaluate the chronic effects of these NPs, especially considering their application in medical imaging and therapy?

4.        The impact of NPs on RBCs is an important aspect of the study. Can the authors discuss whether the observed changes in RBCs could impact their functionality or lifespan?

5.        Were there any observations related to the immune response elicited by these NPs?

6.        The manuscript briefly mentions the potential of these NPs in PDT. Could the authors provide more details on how they envision these NPs being used in PDT?

7.        Are there specific types of tumors or tissues where these NPs might be more effective in imaging and PDT?

Author Response

Response to Reviewer 1 Comments

Point 1: Could the authors provide more detail on the hydrothermal method used for the synthesis of the NPs? Specifically, how does this method compare to other synthesis methods in terms of particle size uniformity and surface properties?

Response 1: We have added the requested information to the manuscript (see below).

“There are several widely used methods for the synthesis of UCNs [15], each of which has a number of unique features that determine its advantages and disadvantages.

The thermal decomposition method allows the synthesis of particles with a high degree of monodispersity with the possibility of particle size control in a relatively short reaction time [15-17]. The main disadvantage of this method is the need to maintain a high reaction temperature (250-330°C) in an oxygen-free environment. This places high demands on the equipment. Toxic wastes are also a major drawback [16]. In addition, the synthesized particles are usually stabilized by a surfactant. This poses difficulties in biological applications and requires surface modification [17].

The co-precipitation method is characterized by the absence of toxic by-products, by the synthesis temperatures that are not so high and by its relative simplicity [15,16]. However, particles with a high degree of monodispersity cannot be obtained with this method. It should also be noted that the resulting particles contain a large amount of adsorbed water. This negatively affects their luminescence properties [15,16].

The sol-gel process is distinguished from other processes for synthesizing UCNs by its higher yields [15, 18]. The inability to control particle size and their significant aggregation are the main disadvantages of this method. Particle size is a critical parameter when using particles for biomedical applications. Particles and particle aggregates that are too large are difficult to remove from the body and can lead to capillary blockage. Particles that are too small have cellular permeability and, as a result, toxicity [19]. Therefore, to produce particles for biomedical applications, the sol-gel method is rarely used to synthesize UCNPs.

One of the most common synthesis methods is solvothermal synthesis [16]. The method is based on the high solubility of inorganic substances at elevated temperatures and pressures and the subsequent growth of crystals from the liquid phase. The necessary components of the system are a solvent, a mineralizer and precursors for the grown crystals. The result of the use of organic solvents is the production of small, homogeneous particles. However, when the synthesis is done in organic solvents, the toxicity of the resulting substances increases the requirements for the equipment and the synthesis conditions.

The hydrothermal method, where water is the solvent, is a special case of solvothermal synthesis. This synthesis method is widely used due to its ease of implementation and low cost of starting materials. At the same time, carrying out the synthesis in an aqueous environment simplifies the process of surface modification for biomedical applications [15, 20].

The hydrothermal method is promising for the production of ACC. It offers a wide range of possibilities for obtaining particles with specific parameters while controlling the influencing synthesis conditions. The data of the scientific groups show that by changing the conditions of the hydrothermal synthesis it is possible to obtain particles of different morphologies, from rods to plates, and of different sizes, from 10 nm to 5 μm [21, 22]. The hydrothermal method seems to be the most suitable because of the absence of toxic synthesis products and the possibility to prepare intensely luminescent particles of given sizes [23]. The issue of hydrothermal synthesis is discussed in more detail in the authors' paper [24].”

Point 2: The manuscript mentions moderate toxicity of the NPs. Could the authors elaborate on the criteria used to determine this level of toxicity?

Response 2: We have added the requested information to the manuscript (see below).

«Pathologic anatomy, by definition, distinguishes between reversible changes (impaired perfusion, edema, infiltration of blood cells) and irreversible changes (necrosis, apoptosis) [35]. Therefore, when describing histologic changes, if we observe mainly signs of impaired blood supply, moderate cell damage, and the absence of necrosis of organ cells, we conclude that the agents under study are moderately toxic.»

Point 3: Is there any long-term study planned or conducted to evaluate the chronic effects of these NPs, especially considering their application in medical imaging and therapy?

Response 3: Currently, the authors of this paper are conducting research on the effect of photodynamic exposure on tumor growth and organ condition with repeated administration of nanoparticles. The team has published a number of papers on the toxicity of synthesized nanoparticles [Yanina I. I., Kochubey V. I. Toxicity of Upconversion Nanoparticles. Overview. Izvestiya of Saratov University. Physics , 2020, vol. 20, iss. 4, pp. 268-277. DOI: 10.18500/1817-3020-2020-20-4-268-277; Sagaidachnaya Е. А., Yanina I. Yu., Kochubey V. I. Prospects For Application of Upconversion Particles NaYF4 : Er,Yb for Phototherapy. Izv. Saratov Univ. (N. S.), Ser. Physics, 2018, vol. 18, iss. 4, pp. 253–274 (in Russian).DOI: https://doi.org/10.18500/1817-3020-2018-18-4-253-274; Irina Yu. Yanina, Elena K. Volkova, Elena A. Sagaidachnaya, Nikita A.Navolokin, Dmitry A. Mudrak, Andrey M. Zakharevich, Vyacheslav I. Kochubey, Valery V. Tuchin, "Interaction of upconversion luminescent nanoparticles with tissues and organs," Proc. SPIE 10685, Biophotonics: Photonic Solutions for Better Health Care VI, 106852X (17 May 2018); doi:10.1117/12.2304709]. We have added the requested information to the manuscript (see below).

“We will take the results obtained into consideration when conducting research on the effect of photodynamic exposure on tumor growth and organ condition with repeated administration of nanoparticles. ”

Point 4. The impact of NPs on RBCs is an important aspect of the study. Can the authors discuss whether the observed changes in RBCs could impact their functionality or lifespan?

Response 4: We have added the requested information to the manuscript (see below).

“The observed changes in the shape of erythrocytes and the appearance of echinocytes contribute to the observed increase in aggregation. However, it was not possible in our study to assess the effect on erythrocyte functionality or lifespan, but no increase in erythrocyte destruction (hemolysis was not detected).”

Point 5. Were there any observations related to the immune response elicited by these NPs?

Response 5: We have added the requested information to the manuscript (see below).

“From the point of view of the immune response to the introduction of nanoparticles, changes in the spleen can be analyzed. No significant changes were observed.”

Point 6. The manuscript briefly mentions the potential of these NPs in PDT. Could the authors provide more details on how they envision these NPs being used in PDT?

Response 6: We have added the requested information to the manuscript (see below).

“In addition, the conditions for PDT can be improved by creating complex NPs with target molecules and a photodynamic dye adsorbed to the surface [9]. The photodynamic dye absorbs the luminescence of the NPs. Thus, compared to direct excitation of the dye in its absorption band, it is possible to act at a greater depth. One of the most promising materials for the development of PDT methods are functionalized UCNPs [10]. UCNPs can not only increase the efficacy of the therapy, but also provide an additional photothermal effect while controlling the temperature of the treatment area [11-13]. For PDT, it is promising to prepare an UCNPs (e.g., NaYF4:Er,Yb) and then coat it with a PS-containing shell [14]. In this case, the efficiency of excitation of PS molecules is determined by the condition of the need to overlap the spectra of upconversion luminescence and the absorption of PS.”

Point 7. Are there specific types of tumors or tissues where these NPs might be more effective in imaging and PDT?

Response 7: We have added the requested information to the Conclusion (see below).

“This type of nanoparticle may be more effective in imaging and PDT for superficial tumors with developed blood vessels. More particles will enter and accumulate. For example, hemangiomas, skin tumors, glioblastomas, meningiomas.”

Dear Editor,

We thank referee for thorough review of our manuscript. We are very thankful for the effort invested by the referee, for the constructive suggestions, and we feel that the quality of our paper has been substantially improved with the corresponding changes. We agree with comments of the reviewer. We did changes suggested by the reviewer. All changes are highlighted in the paper text.

Best regards, Irina Yanina

Reviewer 2 Report

Comments and Suggestions for Authors

1. In SEM, the nanoparticles tend to aggregate, the author should add DLS data to prove the nanoparticles are separated in solution.

2. The author should add the stability experiment to prove the nanoparticles can be stable in physiological conditions for a long time. 

3. There is no Figure 3 and double Figure 5 in the manuscript.

Comments on the Quality of English Language

There are some spelling errors, the authors should check carefully.

Author Response

Point 1: In SEM, the nanoparticles tend to aggregate, the author should add DLS data to prove the nanoparticles are separated in solution.

Response 1: We have added the requested information to the Supplementary Materials of manuscript (see below).

Figure S3-3. Data of DTS for synthesized NPs. Average size of NPs is 53 nm

Point 2: The author should add the stability experiment to prove the nanoparticles can be stable in physiological conditions for a long time

Response 2: We have added the requested information to the Supplementary Materials of manuscript (see below).

“Synthesized particles can be stored in physiological solution at room temperature for more than one month without any change in luminescence intensity.”

Point 3: There is no Figure 3 and double Figure 5 in the manuscript.

Response 3: We have taken into account the reviewer's comment. (see below). We have corrected the numbering of images.

Point 4. There are some spelling errors, the authors should check carefully.

Response 4: Thank you, fixed.

Reviewer 3 Report

Comments and Suggestions for Authors

In this manuscript, Ana Doronkina et al. investigated NaYF4 doped with Yb and Er UC nanoparticles for imaging applications. The authors did a very interesting study, presenting the literature background, methods, results, and conclusions. I recommend this manuscript be published in your journal.

However, only a several things should be improved.

Regarding SEM images, can you take some micrographs with higher magnification? It would be useful in order to see the shape of smaller particles.

In the PL spectrum, please mark the appropriate transitions. Why is the intensity ratio of green and red emissions changing? Give a more detailed explanation.

Comments on the Quality of English Language

Only a few English mistakes were found in the text. 

Author Response

Point 1: Regarding SEM images, can you take some micrographs with higher magnification? It would be useful in order to see the shape of smaller particles.

Response 1: We have added the requested information to the manuscript (see below).

 (a)   (b)

Figure 1. SEM images of NaYF4: Yb3+, Er3+ particles coated with SiO2 shell. Magnification 30x (a) and 200x (b)

Point 2: In the PL spectrum, please mark the appropriate transitions. Why is the intensity ratio of green and red emissions changing? Give a more detailed explanation.

Response 2: We have added the requested information to the manuscript (see below).

“The erbium radiative transitions that form the luminescence spectrum are shown in Table 1. The lines show a fine structure due to the interaction with the hexagonal lattice of the NPs. The relative decrease in intensity of the red radiation is explained by its greater absorption by Methylene Blue, which is located in the shell of the nanoparticles. ”

Table 1. Erbium Radiative Transitions.

522 nm

541 nm

658 nm

2H11/2 (Er) ® 4I15/2 (Er)

4S3/2 (Er) ® 4I15/2 (Er)

4F9/2 (Er) ® 4I15/2 (Er)

Point 3: Only a few English mistakes were found in the text.

Response 3: Thank you, fixed.

Reviewer 4 Report

Comments and Suggestions for Authors

Dear Authors,

The main question addressed in this research is clearly stated and well-defined. It provides a good foundation for the entire study, making it easy for readers to understand the focus of the investigation.

The topic addressed in this article is relevant to the field, and the research seems to fill a specific gap in the existing literature.

However, it would be beneficial if the authors could explicitly highlight the unique aspects that make their study original compared to previous works.

The article effectively builds upon existing literature, and the findings contribute valuable insights to the subject area. However, a more detailed discussion on how this research extends or challenges previous studies would enhance the manuscript's impact.

While the methodology is generally sound, there is a need for a more comprehensive discussion on potential limitations and uncertainties. Consider providing additional details on the luminescence spectrum of NP.  Additionally, a clearer description in fig2 of the difference between the two spectrum would enhance the reader's understanding. In general paragraph 3.2 must be improved from the point of view of clarity.

The conclusions drawn in the article are consistent with the evidence presented. However, the authors should address any discrepancies or unexpected results, providing a more nuanced interpretation where needed. Additionally, a stronger link between the conclusions and the use of NP for imaging would strengthen the overall argument.

The references cited are generally appropriate and cover relevant literature. However, it would be beneficial to include more recent studies if available, especially if they are directly related to the research topic.

The tables and figures are well-presented and contribute effectively to the understanding of the research. However, consider providing more detailed captions that explain the key takeaways from each table or figure. Additionally, clarity could be improved in certain figures. Where fig 3 is? I suppose the Fig3 is Fig 4 and so on.

Author Response

Point 1: However, it would be beneficial if the authors could explicitly highlight the unique aspects that make their study original compared to previous works.

Response 1: The main uniqueness is the complexity of the research. Most articles deal separately with imaging, the effect of nanoparticles on red blood cells, or histological studies. This paper combines these aspects. We have added the requested information to the Introduction section.

“ That is, extensive research on nanoparticle effects on red blood cells and histological analysis of organs and tissues.”

Point 2: The article effectively builds upon existing literature, and the findings contribute valuable insights to the subject area. However, a more detailed discussion on how this research extends or challenges previous studies would enhance the manuscript's impact.

Response 2: As mentioned above, the main advantage of this article is the complexity of the research. In addition, research on the effect of NaYF4 nanoparticles on erythrocytes is practically absent in the literature. There are quite a few studies on gold, magnetic nanoparticles and nanodiamonds, including a very good review article [de la Harpe, K.M.; Kondiah, P.P.D.; Choonara, Y.E.; Marimuthu, T.; du Toit, L.C.; Pillay, V. The Hemocompatibility of Nanoparticles: A review of cell-nanoparticle interactions and hemostasis. Cells 2019, 8, 1209. https://doi.org/10.3390/cells8101209], but we did not find any studies similar to ours.We have taken into account the reviewer's comment (see below).

“ In a review article, Mehrizi T.Z. [29] examined the data on the hemocompatibility of NPs with RBC for the period from 2010 to 2020. The reviewed literature shows that negatively charged dendrimers, unsaturated/uncharged liposomes, pegylated NPs and RBC can improve RBC quality and hemocompatibility. However, large cationic dendrimers, liposomes composed of saturated long acyl chain lipids and cationic chitosan NPs are less compatible with RBC. In addition, these polymeric NPs allow for more surface modifications and manipulations, making them more hemocompatible. Among metal NPs, gold and iron NPs have reasonable compatibility with RBC. However, the smaller size, higher concentration and longer exposure time of these NPs may induce hemolysis and morphological changes in RBC.

When NPs larger than 50 nm are used, the main mechanism of interaction with RBC is membrane interaction. This changes the shape and size of the RBC and the degree of aggregation. In numerous and varied experiments with blood samples, it has been shown that the degree and rate of erythrocyte aggregation depends not only on the concentration of the cells, the state of their surface, but also on the physicochemical properties and concentration of blood plasma proteins [30, 31].

However, one must ensure that the binding of NPs to the RBC membrane does not lead to disastrous morphological and physiological consequences. It is necessary to determine the effect on blood cells before injecting the drug into the bloodstream.”

Point 3: While the methodology is generally sound, there is a need for a more comprehensive discussion on potential limitations and uncertainties. Consider providing additional details on the luminescence spectrum of NP. Additionally, a clearer description in fig2 of the difference between the two spectrum would enhance the reader's understanding.

Response 3: The characteristics of the multiphoton processes leading to the appearance of upconversion luminescence are beyond the scope of this article. The main advantage of upconversion luminescence is described in the introduction [Zhang F. Photon upconversion nanomaterials. Springer, 2016. 428 p. DOI:10.1007/978-3-662-45597-5]. We have added the requested information to the text.

“ The erbium radiative transitions that form the luminescence spectrum are shown in Table 1. The lines show a fine structure due to the interaction with the hexagonal lattice of the NPs. The relative decrease in intensity of the red radiation is explained by its greater absorption by Methylene Blue, which is located in the shell of the nanoparticles. ”

Table 1. Erbium Radiative Transitions.

522 nm

541 nm

658 nm

2H11/2 (Er) ® 4I15/2 (Er)

4S3/2 (Er) ® 4I15/2 (Er)

4F9/2 (Er) ® 4I15/2 (Er)

Point 4. In general paragraph 3.2 must be improved from the point of view of clarity.

Response 4: At this point, we wanted to show that the nanoparticles we used can be used to irradiate a luminescent region with a broad beam and photograph it. This is in contrast to the widely used method of raster point-by-point registration, where the excitation radiation is scanned over an object and the luminescence is recorded at maximum flux collection. The ability to visualize depends on the quality of the synthesized nanoparticles, in particular the quantum yield of the luminescence. The data presented show that these nanoparticles are quite suitable for visualization, but it is necessary to study the effect on organs (toxicity) and cellular elements. This is described in the article. We have taken into account the reviewer's comment (see below).

3.2 Imaging

To localise the NPs precisely, it is necessary to image where the NPs lie within the tumour. This allows the NPs to be irradiated directly in the tumour area without irradiating neighbouring areas. It should be noted that we did not perform the visualisation in a point-by-point mode, but by photoregistration of the entire image within the irradiation area (1 cm).

3.3 Histology…

Point 5. The conclusions drawn in the article are consistent with the evidence presented. However, the authors should address any discrepancies or unexpected results, providing a more nuanced interpretation where needed. Additionally, a stronger link between the conclusions and the use of NP for imaging would strengthen the overall argument.

Response 5: We have taken into account the reviewer's comment. (see below).

“ We have shown that it is possible to obtain luminescent images of the localisation site of the nanoparticles we use by excitation of the luminescent region with a broad beam and photoregistration. The possibility of imaging depends in particular on the quality of the synthesised nanoparticles, in particular on the quantum yield of luminescence. In order to precisely locate the nanoparticles, it is necessary to image where the nanoparticles are located within the tumour. This allows the nanoparticles to be irradiated directly in the tumour area without irradiating neighbouring areas. It should be noted that we did not perform the visualisation in a point-by-point mode, but by photoregistering the entire image within the irradiation area (1 cm).”

Point 6. The references cited are generally appropriate and cover relevant literature. However, it would be beneficial to include more recent studies if available, especially if they are directly related to the research topic.

Response 6: We have taken into account the reviewer's comment (see below).

Point 7. The tables and figures are well-presented and contribute effectively to the understanding of the research. However, consider providing more detailed captions that explain the key takeaways from each table or figure. Additionally, clarity could be improved in certain figures. Where fig 3 is? I suppose the Fig3 is Fig 4 and so on.

Response 7: We have taken into account the reviewer's comment (see below). We have corrected the numbering of images.

Round 2

Reviewer 1 Report

Comments and Suggestions for Authors

This work can be accepted

Reviewer 2 Report

Comments and Suggestions for Authors

it can be accepted in current version